# Degradation of Organic Methyl Orange (MO) Dye Using a Photocatalyzed Non-Ferrous Fenton Reaction

**DOI:** 10.3390/nano13040639

**Published:** 2023-02-06

**Authors:** Sifani Zavahir, Tasneem Elmakki, Nourhan Ismail, Mona Gulied, Hyunwoong Park, Dong Suk Han

**Affiliations:** 1Center for Advanced Materials (CAM), Qatar University, Doha P.O. Box 2713, Qatar; 2School of Energy Engineering, Kyungpook National University, Daegu 41566, Republic of Korea; 3Department of Chemical Engineering, College of Engineering, Qatar University, Doha P.O. Box 2713, Qatar

**Keywords:** methyl orange, titanium nanotube, non-ferrous Fenton reaction, advanced oxidation process, photocatalysis

## Abstract

Removal of recalcitrant organic pollutants by degradation or mineralization from industrial waste streams is continuously being explored to find viable options to apply on the commercial scale. Herein, we propose a titanium nanotube array (based on a non-ferrous Fenton system) for the successful degradation of a model contaminant azo dye, methyl orange, under simulated solar illumination. Titanium nanotube arrays were synthesized by anodizing a titanium film in an electrolyte medium containing water and ethylene glycol. Characterization by SEM, XRD, and profilometry confirmed uniformly distributed tubular arrays with 100 nm width and 400 nm length. The non-ferrous Fenton performance of the titanium nanotube array in a minimal concentration of H_2_O_2_ showed remarkable degradation kinetics, with a 99.7% reduction in methyl orange dye concentration after a 60 min reaction time when illuminated with simulated solar light (100 mW cm^−2^, AM 1.5G). The pseudo-first-order rate constant was 0.407 µmol^−1^ min^−1^, adhering to the Langmuir–Hinshelwood model. Reaction product analyses by TOC and LC/MS/MS confirmed that the methyl orange was partially fragmented, while the rest was mineralized. The facile withdrawal and regeneration observed in the film-based titanium nanotube array photocatalyst highlight its potential to treat real industrial wastewater streams with a <5% performance drop over 20 reaction cycles.

## 1. Introduction

Advanced oxidation processes (AOPs) are an emerging method of treating organic contaminants in wastewater, characterized by the in situ generation of highly active oxidizing free radicals [1]. Free radical formation can be induced by many media such as H_2_O_2_ [2], O_3_ [3], Cl_2_ [4], UV–Visible radiation [5], heat, persulfates and peroxymonosulfates [6], carbonates [7], and nitrate–nitrite mixtures [8]. Photocatalysis-based AOPs are driven by Fenton or Fenton-like chemistry [9], ozonation [10], and sulfate radical-based processes [11]. Herein, Fenton processes generate in situ hydroxyl radicals at a redox potential of 2.8 V, non-selectively attacking a wide array of chemical centers, leading to fragmentation and even mineralization of the parent molecules [12]. The classical Fenton process involves Fe(II) species that react with H_2_O_2_ to produce ·OH radicals in situ. However, this process has limitations, primarily due to the precipitation and sludge formation of Fe(III), with the former process reducing the available Fe(II) content in the system over time and significantly lowering the reaction efficiency. The replacement of Fe(II) with other metals, alloys, and metal oxides for organic substance decomposition through the subsequent generation of ·OH radicals has recently gained much attention [13,14].

Unlike polymer composites for treating aqueous waste streams [15], metals with variable oxidation states act as promising non-ferrous Fenton catalysts in hydrogen peroxide solution due to their multiple oxidation states, which can be catalytically converted from inert to active or vice versa via a simple redox cycle [16]. In this regard, zero-valent aluminum in an acidic medium (pH ˂ 4) has demonstrated Fenton-like activity that mineralizes phenol, 4-chlorophenol, and nitrobenzene [17]. This study found that the decomposition of bisphenol-A was achieved with an efficiency of 75% within 12 h in a highly acidic medium (pH~1.5) and that the rapid formation of reactive OH radicals caused by the addition of trace amounts of Fe(II) increased the decomposition rate exponentially [18]. Another study using a Ce^3+^/H_2_O_2_ system was found to act as an alternative to the Fe^2+^/H_2_O_2_ system, but the extent of ·OH radical generation was highly dependent on the surface defects and surface oxide states of CeO_2_ [19,20]. The degradation kinetics of the acid orange 7 dye compound indicated that the oxidation proceeds through an intermediate hydroxyl adduct followed by adsorption of the dye to the CeO_2_ surface [20]. However, the acute cytotoxicity of Ce in humans does not favor the use of Ce in large-scale applications [21]. Unlike most non-ferrous metals for the Fenton process, the Cr(VI) species has been found to generate highly active Cr(V) during the in situ formation of stable OH radicals over a wide pH range [22]. The rapid dissolution of Cr(VI) and extreme carcinogenic properties of Cr(VI) in aqueous media hinder its use in water treatment, as Cr(VI) is released after treatment, which opens up another threat that requires post-treatment again [23,24]. Furthermore, Co [25,26], Mn [27], Cu [28,29], and Ru [30,31] species have been studied in non-ferrous Fenton processes in mild acid, neutral, or weakly alkaline pH ranges.

TiO_2_ has been used in UV-activated AOP systems with ozone, where hydroxyl radicals were produced by ozone decomposition in water [32,33]. The resulting system was highly active in decomposing recalcitrant 2,4-dichlorophenoxyacetic acid, bisphenol A, butylnaphthalenesulfonate, and benzyldodecyldimethyl ammonium bromide, leading to efficient mineralization as seen in total organic carbon (TOC) measurements [32]. Herein, TiO_2_ is non-toxic, but using ozone makes the process less environmentally benign [34]. The rapid reactivity of ozone induces undesirable side reactions that generally lead to toxic byproducts. Hydroxide radicals generated from H_2_O_2_ are generally short-lived and formed under normal temperature and pressure conditions [35], so there are no major limitations on reactor design and apparatus. Thus, a simple and uncomplicated reactor design is sufficient to operate as a stand-alone single system or to work with another treatment method in a hybrid process. Hence, a TiO_2_/H_2_O_2_ system replacing ozone would be very attractive if it can produce surface active radicals to drive AOPs. Recently, Choi et al. showed that inorganic As(III) was oxidized to As(V) in a TiO_2_/H_2_O_2_ system along with WO_3_/H_2_O_2_ and ZrO_2_/H_2_O_2_ [36]. The authors suggested that the reaction proceeds through surface complexation between TiO_2_ and H_2_O_2_ to form hydroperoxyl radicals on the surface of TiO_2_ by an inner sphere electron transfer. In support of this, the authors found that the rate of As(III) to As(V) conversion was least affected by hydroxyl radical scavengers, while there was a significant decrease in the reaction rate for hydroperoxyl or superoxide radical scavengers. Recently, a N-TiO_2_/H_2_O_2_/visible light system was evaluated for the degradation of diclofenac in a submerged photocatalytic membrane reactor [37]. Even when the reaction proceeded with high mineralizing efficiency, the authors found that recovering the suspended catalyst powders was costly and laborious. This inherent downfall can be easily overcome by using a film-based catalyst that can easily enable the scalability of the reaction system.

As discussed above, AOPs are indispensable for treating organic or inorganic contaminants in ferrous or non-ferrous Fenton-type processes. Regardless of the type, most work has focused on catalysts in powder form. Catalysts grown on their own in a film undoubtedly have the highest level of stability that can be easily separated from the reaction system. However, the summary provided in Table 1 clearly indicates that a suitable catalytic system for treating common industrial pollutants in a process that can be easily operated utilizing abundant sunlight is still to be improved from a pilot-scale system to an industrial-scale system.

It is known that TiO_2_ can be synthesized in the form of nanotubular arrays on Ti film by an anodization process, rendering fewer interfacial grain boundaries [38] to improve charge separation and enhance redox activity. In this study, the non-ferrous Fenton behavior of TiO_2_ was explored with a vision to be tested on a pilot scale in the future, with the addition of a small amount of H_2_O_2_ under light irradiation for the degradation of methyl orange (MO), one of the major organic contaminants. Therefore, in this study, TiO_2_ nanotube arrays were fabricated and systematically utilized with a non-ferrous Fenton approach with pharmaceutical grade 6% H_2_O_2_ to degrade the model contaminant, MO, under natural sunlight irradiation. The catalysts were characterized by surface analysis techniques such as XRD, XPS, SEM, and profilometry. The MO degradation rate was studied as a function of H_2_O_2_ concentration, the intensity of solar light, and the type and composition of TiO_2_ material. A plausible reaction mechanism was derived from controlled experiments using hydroxyl and hydroperoxyl radical scavengers. In addition, the catalyst was reused for 20 cycles to compare the recyclability of the TiO_2_ film catalyst for each MO degradation.

## 2. Experimental Methods

### 2.1. Materials

Phosphoric acid (H_3_PO_4_), acetone, ethanol, sodium fluoride (NaF), ethylene glycol (EG), methyl orange dye (MO), and H_2_SO_4_ (98%) were purchased from Sigma Aldrich (St. Louis, MO, USA) in the highest purity grade available and used as received. Titanium sheets (99.7% purity, Yunjie Metal Co., Baoji, China), stainless steel bars (Yunjie Metal Co., China), and polishing cloth (Struers LLC, Cleveland, OH, USA) were used to fabricate titania nanotube arrays. The H_2_O_2_ used in the experiments was pharmaceutical grade with 6% solution in water (Meliorate Health, Mumbai, India). The water used in all experiments was deionized ultrapure water (DI), attained from a Milli-Q direct water purification system (Fischer Scientific, Waltham, MA, USA) with a resistance of 18.2 MΩ at 25 °C.

### 2.2. Catalyst Film Preparation

Titania nanotube array (TNA) films were fabricated using a modified anodization technique stated elsewhere [39,40]. In short, titanium sheets were cut into small pieces of 2 × 4 cm^2^ and then polished with sandpaper (400 grit) and a wet polishing cloth with an alumina micro polish slurry to create a flat, defect-free surface and remove the oxide layer on the surface. The polished Ti film was then ultrasonically cleaned in acetone and ethanol for 10 min each, then completely dried under a N_2_ gas stream. Next, the cleaned Ti film was electrochemically anodized in an electrolyte solution containing 4:1 (*v*/*v*) of DI: EG, 0.26 M of NaF, and 0.94 M H_3_PO_4_. In the electrochemical cell, the Ti film was used as the anode and paired with a stainless-steel sheet as the cathode, and the distance between the two electrodes was less than 3 cm. After the electrochemical cell was set up, a direct current (DC) voltage of +30 V and 0.04 A was applied to the cell for 4 h and continuous stirring of the electrolyte was ensured during the anodization period to provide proper mass transport of charged ion particles in solution. The as-anodized Ti nanotube array film (anoTNA) was washed thoroughly with deionized water and ethanol, dried in ambient air, and annealed at 400 °C for 2 h to crystallize the anatase phase, and the resulting film was labeled as annTNA.

### 2.3. Dye Degradation Tests

A clean, dry reactor with a quartz window was used for MO dye degradation tests. In a typical reaction, the total liquid volume was 45 mL, containing 20 ppm (61 µM) MO concentration and 1 mL of 6% H_2_O_2_, with 2 mM H_2_SO_4_ acid used to maintain a medium pH of 3.0. The protonation or deprotonation of the dye is important depending on the mechanistic pathway of degradation and affect the adsorption capacity of the dye to the catalyst [41]. A 2 × 1 cm^2^ area of the annTNA catalyst film was immersed in the solution facing the quartz window to maximize light absorption. In experiments studying the H_2_O_2_ effect, the amount of H_2_O_2_ was varied appropriately. Radical scavengers such as isopropyl alcohol (IPA), formic acid (FA), tertiary butyl alcohol (TBA), and ascorbic acid (AA) were used in an amount equivalent to that of the H_2_O_2_ in the medium. In a comparative study with other types of titania, 200 mg of powdered TiO_2_ catalyst was used, which is equivalent to the Ti content of the immersed annTNA film catalyst. The photoelectrochemical reaction was carried out for up to 60 min at a light intensity of 100 mW cm^−2^ irradiated with a solar simulator (ABET Technologies, Milford, CT, USA) equipped with a Xe Arc lamp. Test aliquots of 1 mL were collected at regular intervals, diluted with 2 mL of DI, and tested directly using a UV–Visible spectrometer (Biochrom, Cambridge, UK).

### 2.4. Characterization of Solid and Liquid Samples

The surface morphology and crystalline pattern of the synthesized TNA films were analyzed using scanning electron microscopy with energy dispersive X-ray spectroscopy (SEM/EDX, NOVANANOSEM 450, FEI company, OR, USA) and X-ray diffraction (XRD, PANalytical Empyrean, Malvern Panalytical, Malvern, UK), respectively. Additionally, the surface roughness and depth of numerous photocatalyst films were investigated using a profilometer (Leica DMC 8, Leica Microsystems, Wetzlar, Germany). X-ray photoelectron spectroscopy (XPS) (Kratos, Axis Ultra DLD, Kratos Analytical Ltd, Manchester, UK) was used to collect elemental information and oxidation states of electrode samples in Al Kα monochromator mode. The photoexcitation behavior of annTNA was assessed by photoluminescence (PL) data recorded on an F-7000 FL (Hitachi, Tokyo, Japan) spectrometer with excitation at 375 nm, and emission was measured in the 400–800 nm range. After degradation testing, the degradation products of the MO dye were examined by LC/MS analysis using an Agilent 6460 (Agilent Scientific Instruments, Santa Clara, CA, USA) LC/MS/MS tandem mass unit with an electrospray ionization detector in negative mode. High-performance liquid chromatography (HPLC) analysis was performed on a Waters Acquity (Agilent Scientific Instruments, Santa Clara, CA, USA) instrument equipped with an RP C18 column and a photodiode array (PDA) detector for parallel recognition of MO dye degradation products.

## 3. Results and Discussion

### 3.1. Structural Characterization of the Prepared TNA Films

The annealed titania nanotube array films (annTNA) synthesized in this study showed very regular nanotube arrays with an inner tube diameter of ~100 nm, as shown in Figure 1a. To visualize the average tube length and the alignment of the tubes, the SEM sample specimen was scratched purposefully. As given in Figure 1b,c, annTNA tubes were regularly aligned with similar tube lengths of ~400 nm. The shape of the tube did not change at all from the anodizing process to the annealing step, so the tubular morphology was intact during the crystallization process. XRD analysis showed similar diffractograms for the pure Ti film and the anodized TNA film. Given that the photograph of the anodized TNA film (Figure 1g) shows full coverage of the TNA material on the Ti film, it is less likely that the diffraction pattern seen on the anodized TNA comes from the substrate Ti metal. Since the d-spacing of the TiO_2_ rutile phase coincides with that of the Ti metal [42], it is reasonable to assume that the TNAs formed on the anodized Ti film were a rutile phase, which was partially transformed into an anatase phase in the annealing step. This was confirmed by the diffractogram of annTNA, that exhibited new peaks at 25.30°, 38.57°, and 48.02° related to (101), (004), and (200) planes of the anatase phase, as shown in Figure 1d. XPS analysis of anoTNA film and its annealed TNA counterpart showed no differences in the binding energies of the Ti2p_3/2_ and O1s. The highest peaks of the Ti2p_3/2_ and O1s XPS spectra were centered at 459.3 eV and 530.7 eV, respectively (Figure 1e,f).

### 3.2. Photocatalytic Degradation of MO Dye

The validity of the hypothesis that the titanium center of the annTNA arrays synergizes with H_2_O_2_ to generate oxidative radicals capable of decomposing organic pollutants was examined by the photocatalytic efficiency of annTNA in degrading 20 ppm MO dye using a 6% H_2_O_2_ solution under solar irradiation (AM 1.5 G, 100 mW cm^−2^). The degradation kinetics of the MO dye were also compared to those of the individual H_2_O_2_ and annTNA counterparts over 1 h, with measurements taken at regular time intervals. Figure 2a–c shows that the annTNA–H_2_O_2_ coupled system degraded more than 99% of the dye molecules within 1 h of light irradiation, with 94% MO degradation after 40 min of reaction. The dependence of the MO dye concentration on the resulting degradation rate was thoroughly analyzed with zero-, half-, first- and second-order rate equations. Consistent with several reported works in the literature, the kinetics of MO dye degradation by the annTNA–H_2_O_2_ coupled system followed the Langmuir–Hinshelwood (L–H) reaction kinetics. Several reaction parameters govern the L–H kinetics, including the incident light intensity (*I*), the light-exposed surface area of the catalyst (*s*), the concentration of the MO dye (C), and the adsorption coefficient of the dye on the catalyst surface (*K*). Considering the effect of the parameters specified above, the reaction rate (Γ) can be written as in Equation (1).
(1)$$\Gamma =-\frac{\mathrm{dC}}{\mathrm{dt}}=\;f\left(I,\;s,\;C,\;K\right)$$

More specifically, the photocatalytic degradation of the MO dye, which proceeded through the L–H mechanism, follows a pseudo-first-order reaction rate where a single molecule of MO dye is attached to the active site of the TNA photocatalyst per occurrence. It can be expressed as Equation (2):(2)$$\Gamma =-\frac{\mathrm{dC}}{\mathrm{dt}}=k\frac{KC}{1+KC}$$
where t is the reaction time and *k* is the L–H rate constant. MO dye concentrations over time were fitted using the nonlinear regression function, ‘nlinfit’, in MATLAB. The resulting L–H rate constant value (*k*) indicates how fast the dye molecule decomposes, which was dependent on the band gap energy of the photocatalyst, the accessible active radical species, such as hydroxyl, hydroperoxyl, or superoxide radicals, and the lattice oxygen defects. In the photo-irradiation process, the first step in the degradation pathway involves the photo-interaction of titanium centers in TNA, which produces charge carrier pairs within the moiety when the photon energy of incident light is higher than the bandgap of the titania phase (R-3). In the next step, excited titania centers abstract electrons in H_2_O_2_ to form hydroxyl radicals (·OH) and hydroxyl anions (R-4). During the degradation/mineralization process, a radical attack on the dye center is more subtle and facile than an expedition by positive or negative ions. Hydrogen peroxide, known as an oxidant by itself [43], degraded the MO dye by only 45% after 1 h of reaction under sunlight illumination. H_2_O_2_ is known to produce ·OH radicals when exposed to photon energies below 380 nm (R-5). However, natural sunlight (or simulated) emits about 7% UV radiation in its spectrum, which explains the underlying reason for the weak but possible degradation activity. It took 6 h to degrade 88% of the MO dye, and prolonging the reaction time did not increase the degradation rate further. The annTNA alone showed a degradation rate of 10% after a reaction time of 1 h. In a light-irradiated TiO_2_-only system, hydroxyl radicals can be produced from water molecules in aqueous media, as represented in R-6. However, the density of ${\mathrm{TiO}}_{2}\left(h_{\mathrm{VB}\;}^{+}\right)$ in a TiO_2_-only system is low because the anatase-phase TiO_2_ has a bandgap energy of 3.6 eV, lying in the UV region as previously stated, and there is no trigger for charge separation of $\left(h_{\mathrm{VB}\;}^{+}\right)$ and $\left(e_{\mathrm{CB}\;}^{-}\right)$ as in the case of H_2_O_2_ (R-4). The degradation rate of the MO dye by the combined system annTNA + H_2_O_2_ is more than twice that of its components alone, presumably due to a synergistic effect in the oxidative degradation reactions similar to the Fenton process (R-4 and R-6). This makes the TiO_2_ substrate open, available, and accessible to proceed with reactions that continuously form radical species in situ, which is noteworthy for further exploration in this study. The respective dark reaction performed under the same conditions did not show any form of degradation of the dye molecules (Figure 2d).
(3)$${\mathrm{TiO}}_{2}+h\nu \;\to \mathrm{Ti}O_{2}\left(e_{CB}^{-}+h_{VB}^{+}\right)$$
(4)$${\mathrm{TiO}}_{2}\left(e_{CB}^{-}+h_{VB}^{+}\right)+H_{2}O_{2}+h\nu \to {\mathrm{TiO}}_{2}\left(h_{VB}^{+}\right)+{}^{·}\mathrm{OH}+{\mathrm{OH}}^{\_}$$
(5)$$H_{2}O_{2}+\left(h\nu <380\;\mathrm{nm}\right)\to 2{}^{·}\mathrm{OH}$$
(6)$${\mathrm{TiO}}_{2}\left(h_{\mathrm{VB}\;}^{+}\right)+H_{2}O\to {\mathrm{TiO}}_{2}+H^{+}+{}^{·}\mathrm{OH}$$
(7)$${\mathrm{TiO}}_{2}\left(e_{\mathrm{CB}\;}^{-}\right)+O_{2}\to {\mathrm{TiO}}_{2}+{O\;}_{2}^{··˙˙-}$$
(8)$${}^{·}\mathrm{OH}+\mathrm{MO}\;\to \mathrm{MO}{}^{·}+H_{2}O\;\to \;\mathrm{degraded}\;\mathrm{products}$$
(9)$${O\;}_{2}^{··˙˙-}+\mathrm{MO}\;\to \mathrm{MO}{}^{·}+H_{2}O\;\to \;\mathrm{degraded}\;\mathrm{products}$$
(10)$$H_{2}O_{2}+{}^{·}\mathrm{OH}\;\to H_{2}O+{}^{·}{\mathrm{OH}}_{2}$$
(11)$${\mathrm{TiO}}_{2}\left(e_{\mathrm{CB}}^{-}+h_{\mathrm{VB}\;}^{+}\right)\to {\mathrm{TiO}}_{2}+\mathrm{heat}$$

Table 1 compares the annTNA material studied here with other catalytic systems for MO dye degradation. The dye discoloration efficiency by the “annTNA + H_2_O_2_” system strongly displays a substantial enhancement in the MO degradation rate.

**Table 1 nanomaterials-13-00639-t001:** Discoloration efficiency of MO dye by different catalyst materials.

Entry	Catalyst		Experimental	Initial	Decoloration		Ref.
	Material	Form	Conditions	[MO]/	Efficiency/	Time/	
				ppm	%	min	
1	TiO_2_	film	external bias: 0.0 V vs. SCE	20	20	180	[38]
			Arc lamp, 165 mW/cm^2^				
			pH~6.0				
2	CdS/g-C_3_N_4_	powder	0.3 g/L catalyst	2	40	60	[44]
			W lamp, 100 W				
			pH 3.4				
3	ozone	gas	O_3_ gas concentration 68.8 mg/L	400	>99	20	[45]
			ultrasound irradiation 20 kHz				
			heat				
4	Ag-P25	film	UV-A irradiation system,	6	>99	120	[46]
			350 nm, 71.7 µW/cm^2^				
			pH 9.2				
5	Ag_2_CrO_4_/SnS_2_	powder	50 mg catalyst	10	71	120	[47]
			Xe lamp, 500 W				
			pH not given				
6	Fe from steel	suspension	200 mg/L catalyst	20	98	30	[48]
	Industry waste		34 mM H_2_O_2_				
			pH 2				
7	CuO on	powder	0.1 g/L catalyst	10	60	120	[49]
	nanosized		Hg lamp, medium pressure				
	zeolite-X		pH 6				
8	B-doped g-C_3_N_4_	powder	200 mg catalyst	4	70	300	[50]
			Xe lamp, 300 W				
			pH 6.8				
9	g-C_3_N_4_	powder	200 mg catalyst	4	95	300	[50]
			Xe lamp, 300 W				
			pH 6.8				
10	Au-TiO_2_	powder	20 mg catalyst	10	95	160	[51]
			W lamp, 500 W, 300 k lux				
			pH not given				
11	Ag–TiO_2_ porous	powder	50 mg catalyst	100	81.4	180	[52]
	polymer		Xe light, 100 mW/cm^2^				
			pH not given				
12	MoS_2_/YVO_4_	powder	100 mg catalyst	10	98	60	[53]
			Xe lamp, 300 W				
			pH not given				
13	cellulose–TiO_2_	sheet	2.5 cm × 2.5 cm film UV, 100 W lamp	5	95	150	[54]
			pH 3				
14	80% BiOCl/BiOI	powder	20 mg catalyst	50	75	375	[55]
			Xe lamp, 300 W pH not given				
15	Cu(OH)_2_–ZnO_2_	film	1.77 cm^2^ film	8	43	360	[56]
			Hg lamp, 125 W				
			pH not given				
16	ZnO NPs	powder	50 mg catalyst	15	85	180	[57]
			UV lamp, 10 W bulb				
			pH not given				
17	Nb_2_O_5_	powder	5 mg catalyst	20	99.9	20	[58]
			UV lamp				
			pH not given				
18	TiO_2_–PLA	powder	50 mg catalyst	320	81	300	[59]
			UV lamp, 350 nm				
			pH 8				
19	TiO_2_	film	55 mM H_2_O_2_	20	>99	60	**This**
			Xe lamp, 100 mW/cm^2^				**work**
			pH 3.4				

### 3.3. Effect of H_2_O_2_ Concentration

In the annTNA–H_2_O_2_ system, continuously and in situ generated hydroxyl radicals had a crucial role in MO dye degradation, similar to their role in a classical Fenton reaction, where Fe(II) is oxidized to Fe(III), generating ·OH. Hence, the MO degradation kinetics were evaluated over a range of H_2_O_2_ concentrations. Keeping all other experimental parameters constant, increasing the concentration of the H_2_O_2_ from 28 mM to 55 mM showed an exponential increase in the MO degradation rate. As shown in Figure 3a, the highest degradation of 99.7% was observed at 55 mM H_2_O_2_, further increasing the H_2_O_2_ concentration to 82 mM and 110 mM had a detrimental effect on the overall dye degradation rate. It has previously been observed that exceeding a certain concentration of H_2_O_2_ reduces the oxidative power of the whole system as it does not participate in the generation of ·OH radicals and consumes them on their own, as the H_2_O_2_ and MO dye compete with available radicals in the medium (R-9) [60,61,62].

### 3.4. Effect of Radical Scavengers

Most H_2_O_2_-catalyzed reactions are driven by ·OH radicals generated in situ. However, this is not the only possibility, and in certain multi-component systems, the reaction is driven by superoxide radicals (O_2_**·^−^**). Hence, to gain an insight into the MO degradation mechanism in this study, the photocatalytic reaction was performed under optimized conditions (20 ppm MO and 55 mM H_2_O_2_) in the presence of both hydroxyl and superoxide radical scavengers. Isopropyl alcohol (IPA), formic acid (FA), and tert-butanol (TBA) are known as ·OH radical scavengers [36], and ascorbic acid (AA) is known to quench superoxide radicals (O_2_**·^−^**). The MO dye degradation rates shown in Figure 3b were affected by all scavengers used regardless of their type. However, the reduction in the MO degradation rate was much greater with the hydroxyl radical scavengers, and it declined by more than 70% compared to the original MO degradation rate (without scavengers). The MO degradation rates were 20, 26.9, and 21% in the systems with IPA, FA, and TBA, respectively, whereas the rate reduction was relatively low at 79.2% in the system with AA. The experimental results are consistent with the known fact that H_2_O_2_ acts as a precursor to the formation of **·**OH and O_2_**·**^−^, clearly demonstrating that the present degradation reaction is mainly driven by **·**OH, while O_2_**·^−^** affects the reaction insignificantly. Zhao et al. found that the surface complexation of TiO_2_ and H_2_O_2_ leads to the formation of (Ti(IV)-OOH), resulting in **·**OH formation under visible light irradiation [63]. Kim et al. employed a TNA film catalyst combined with H_2_O_2_ to oxidize As(III) to As(V) [36]. Their observations with radical scavengers contrasted with those observed in the current MO dye degradation system, where hydroxyl radical scavengers did not affect the overall reaction rate, while the oxidation of As(III) was significantly reduced in the presence of superoxide radical scavengers. Many studies have shown that most of the oxidation reactions of TiO_2_-based photocatalysts are induced by superoxide radicals when anatase is the only phase present in the TiO_2_ catalyst [64,65]. However, the presence of trace amounts of the rutile phase has been shown to deviate from the conversion of superoxide radicals from H_2_O_2_ to hydroxyl radicals [66,67,68].

### 3.5. Effect of Titania Type

In this regard, other forms and polymorphs of titania were also evaluated for their ability to catalyze the oxidation process synergistically. P25, P90, titanate nanotubes (TNT, synthesized in the lab) powders, and as-anodized TNA (anoTNA) were employed in the MO degradation system under otherwise optimized conditions, as shown in Figure 3c. A comparison of all titania materials showed that the MO decomposition activity of anoTNA film with rutile only was the lowest, possibly due to the absence of a complete anatase phase. In addition, the degradation rate of MO by P25 was better than P90. P25 is a physical crystalline mixture with an anatase-to-rutile ratio of 75:25, while P90 has a ratio of 10:90, meaning that P90 has a higher rutile content than P25. After 10 min of reaction, P25 showed the highest MO degradation rate, which could be attributed to faster charge carrier separation and coupling with H_2_O_2_ than other titania materials. However, as the reaction time continued, annTNA provided the superior MO degradation kinetics, indicating that the catalytic activity of titania increases with less rutile in the titania material. Romanos et al. modified commercial P25 with well-dispersed Cu nanoparticles, and the system was found to degrade 50% of 12 ppm initial dye after exposure to UV light in the 350–390 nm range for 3 h [69]. In another process, a bulk P25 composite using *Posidonia oceanica* fibers showed 100% degradation of phenol after 4 h of UV light irradiation [70]. It is noteworthy that the annTNT–H_2_O_2_ system outperforms commercial P25 in this study. While anatase plays a crucial role in the light-driven process, the trace presence of rutile increases the abundance of hydroxyl radicals, which are the active attacker. TNT powder shows a very good performance, with an 85% MO degradation rate, second only to annTNA, revealing the importance of morphological effects, especially for regular tubular channels. Therefore, based on the observations of various radical scavengers combined with various titania materials, we make a reasonable conclusion that the annealed TNA films (the active catalyst in this study) contain a trace amount of rutile phase in the main anatase moiety. The L–H rate constant and adsorption coefficient values for the previously discussed systems are given in Table 2. The annTNA system has an L–H rate constant of 0.407 µmol^−1^ min^−1^, which is almost 10 times higher than that of anoTNA, at 0.045 µmol^−1^ min^−1^, while that of TNT is 73% higher than anoTNA, at 0.078 µmol^−1^ min^−1^.

### 3.6. Effect of Light Intensity

As already mentioned for light-induced reactions, light intensity makes a pivotal contribution, as it is directly linked to the number of photons reaching the titania catalytic centers of TNA, influencing charge carrier generation and separation (R-3). The effect of light intensity on the overall MO dye degradation rate in the optimized system was studied at four different light intensities: 45, 60, 75, and 100 mW cm^−2^. As shown in Figure 4a, the light intensity has a positive non-linear relationship with the MO degradation rate. As the light intensity increased to 60, 75, and 100 mW cm^−2^, the MO degradation rate of 17% at 45 mW cm^−2^ was improved to 45, 90, and 99%, respectively.

The photoluminescence (PL) spectra presented in Figure 4b provide meaningful insights into the excitation behavior of the fabricated annTNA film. The measured PL emission upon excitation at 375 nm shows three peaks at 482, 524, and 562 nm, all falling in the visible region. These blue and green level emissions can be partly attributed to defect sites in the annTNA film material arising from oxygen vacancies, which greatly contribute to improving the photoactivity of the annTNA film in the visible region.

### 3.7. Electrochemical Characterization of annTNA Film

The important role of light in the MO degradation reaction was further assessed and affirmed by a set of electrochemical tests employing annTNA as the working electrode in a three-electrode configuration. Charge carrier transport resistance in light and dark-assisted media was studied by electrochemical impedance spectroscopy (EIS) tests performed in 0.1 M Na_2_SO_4_. As shown in Figure 4c, the charge carrier transport resistance was low under the light illumination condition, indicated by a lower arc radius than in the dark (larger arc radius). The charge carrier generation with intermittent light illumination was analyzed by chopped chronoamperometry (Figure 4d). The strong and fast response of annTNA to the incident light was demonstrated by the troughs and crests of the chopped chronoamperogram.

### 3.8. Tentative Reaction Mechanism

Based on the experimental results, it can be suggested that the main cause of MO dye degradation in the annTNA and H_2_O_2_ combined system is the reactive hydroxyl radicals formed repeatedly under light illumination (Figure 5).

Reaction products were also studied by total organic carbon (TOC) analysis on samples collected at regular intervals. As the reaction time increased, the TOC decreased, while the total carbon (TC) content remained consistent, indicating that the MO dye was partially mineralized with fragmentation. During mineralization, carbon components of the dye were transformed into CO_2_ or became bicarbonate (HCO_3_^−^) in an acidic medium (pH 3), which was detected as inorganic carbon during the test. An LC/MS/MS tandem analysis revealed fragmented masses corresponding to m/z peaks at 304, 290, 260, 240, 225, and 201 in the first step, and 289, 275, 240, 256, 225, 185, 137, 121, and 112 in the next step (Appendix A). According to the prior literature [71], fragmentation products related to the breakdown of methyl orange dye molecules were proposed, as shown in Figure 1. Other fragmentation products were associated with impurities in the initial dye and are summarized in Figure 2.

### 3.9. Reusability of annTNA Photocatalyst

The recycling ability of the TNA electrode was studied for 20 repeated MO degradation cycles in an optimized environment. After each experiment, the TNA electrode was gently washed with DI and air-dried overnight to reuse in subsequent cycle experiments. As shown in Figure 6a, there was no significant difference in MO dye degradation rates during the 20 cycles. Furthermore, excellent structural integrity over repeated cycles was best evidenced by the SEM image of the TNA electrode evaluated after repeated degradation cycles, proven by undisturbed tubular arrays (Figure 6b). The stability of the annealed film under the consecutive dye degradation runs was evaluated by profilometry, as shown in Figure 6c,d, and a high stability of the film was observed.

In addition to its high catalytic activity, the use of TNA films has significant advantages in separating and recycling the catalyst from the medium. This is mainly due to the ease of managing the operational conditions. This system operated in an environment close to ambient temperature and pressure, indicating the cost-competitive nature of the system when scaled up and its ability to utilize abundant sunlight for wastewater treatment, particularly of textile waste. The system requires an in depth tolerance analysis [72]. Thus, the developed facile regenerative and self-supporting titania nanotube array–H_2_O_2_ coupling process will be evaluated for other contaminants in similar categories, such as phenol, bisphenol A, methyl blue, SRB dye, etc., to understand the capabilities of the TNA–H_2_O_2_ system. The vision is to upscale the system to an industrial level to reach Technology Readiness Level (TRL)-7 or higher to treat real complex industrial wastewater.

## 4. Conclusions

Herein, we demonstrated the potential of a film-based annTNA and H_2_O_2_ coupled system to degrade methyl orange, an azobenzene-type commercial textile dye, via a non-ferrous Fenton-type advanced oxidation process (AOP) under solar illumination with remarkable reaction kinetics. Methyl orange dye was decomposed by almost 99.7% after 60 min of solar irradiation (100 mW cm^−2^, AM 1.5G). TOC, LC/MS/MS, and UV–Vis spectroscopic analysis of the reaction products confirmed that the photocatalytic degradation proceeded by partial fractionation and mineralization. The experimental data were in good agreement with the Langmuir–Hinshelwood model, following pseudo-first-order reaction kinetics. The simple regeneration ability of the self-supporting catalyst through gentle washing of the film with deionized water and its consistent degradation performance (<5% change) over 20 cycles encourages the use of the system on a pilot scale to treat real wastewater.

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
