# Peer review of "Degradation of Organic Methyl Orange (MO) Dye Using a Photocatalyzed Non-Ferrous Fenton Reaction"

_nanomaterials, 2023, doi:10.3390/nano13040639_

Round 1
Reviewer 1 Report
1. This study seems interesting. The experiments are well presented, and the results have value for practitioners. However, discussion is very limited. The chapter Results and discussion is mainly about results. There is barely any discussion involved.
2. Figure 1---Please add scale bar in the SEM micrograph.
3. Scheme 1---Please change to Figure 5
4. The reference format in the introduction is not correct. Please amend it.
5. Strengthen the abstract section. It is very lengthy in preset form. Remove unnecessary information and add key conclusions of the work in the last two lines.
6. Some leading works regarding “Organic Methyl Orange” should be discussed in the introduction.
---Luo, B.; Wu, C.; Zhang, F.; Wang, T.; Yao, Y. Preparation of Porous Ellipsoidal Bismuth Oxyhalide Microspheres and Their Photocatalytic Performances. Materials 2022, 15, 6035.
---Giziński, D.; Mojsilović, K.; Brudzisz, A.; Tiringer, U.; Vasilić, R.; Taheri, P.; Stępniowski, W.J. Controlling the Morphology of Barrel-Shaped Nanostructures Grown via CuZn Electro-Oxidation. Materials 2022, 15, 3961.
---Theodorakopoulos, G.V.; Katsaros, F.K.; Papageorgiou, S.K.; Beazi-Katsioti, M.; Romanos, G.E. Engineering Commercial TiO2 Powder into Tailored Beads for Efficient Water Purification. Materials 2022, 15, 326.
---Luque-Morales, P.A.; Lopez-Peraza, A.; Nava-Olivas, O.J.; Amaya-Parra, G.; Baez-Lopez, Y.A.; Orozco-Carmona, V.M.; Garrafa-Galvez, H.E.; Chinchillas-Chinchillas, M.d.J. ZnO Semiconductor Nanoparticles and Their Application in Photocatalytic Degradation of Various Organic Dyes. Materials 2021, 14, 7537.
---Qiu, J.-P.; Xie, H.-Q.; Wang, Y.-H.; Yu, L.; Wang, F.-Y.; Chen, H.-S.; Fei, Z.-X.; Bian, C.-Q.; Mao, H.; Lian, J.-B. Facile Synthesis of Uniform Mesoporous Nb2O5 Micro-Flowers for Enhancing Photodegradation of Methyl Orange. Materials 2021, 14, 3783.
---Suner, S.S.; Demirci, S.; Sutekin, D.S.; Yilmaz, S.; Sahiner, N. Thiourea-Isocyanate-Based Covalent Organic Frameworks with Tunable Surface Charge and Surface Area for Methylene Blue and Methyl Orange Removal from Aqueous Media. Micromachines 2022, 13, 938.
---Jiang, D.; Song, X.; Zhang, H.; Yuan, M. Removal of Organic Pollutants with Polylactic Acid-Based Nanofiber Composites. Polymers 2022, 14, 4622.
---Alshaikhi, H.A.; Asiri, A.M.; Alamry, K.A.; Marwani, H.M.; Alfifi, S.Y.; Khan, S.B. Copper Nanoparticles Decorated Alginate/Cobalt-Doped Cerium Oxide Composite Beads for Catalytic Reduction and Photodegradation of Organic Dyes. Polymers 2022, 14, 4458.
---Enesca, A.; Cazan, C. Polymer Composite-Based Materials with Photocatalytic Applications in Wastewater Organic Pollutant Removal: A Mini Review. Polymers 2022, 14, 3291.
7. Discuss the motive behind the work. The clear application of the work should be discussed in the introduction section. From the introduction section application of the work is not clear.
8. There are numerous spelling and grammatical errors. Please revise the manuscript thoroughly. Sentences are also not complete.
9. The novelty of the work should also be discussed in a separate paragraph.
10.Try to make a bridge between current and previously published work and specify the gap area and objective of the work. Add the specific gap observed from the literature at the end of the introduction section.
11.Experimental section needs a clear and concise discussion.
12.Work is presented well but the technical discussion is very poor.
Please finish it.
13.Authors should more clearly emphasize the contribution of this work in relation to the existing solutions in the literature.
14.What is the main difficulty when applying proposed method? Authors should clearly state the limitations of the proposed method in practical applications.
15.Please show some directions for the future study.
16.The abstract is also not sufficiently informative, concise and clear. No any quantitative results. Please amend it.
17.Conclusions must be comprehensive and not written like a report. Please amend it.
18.Please add the applicability of present work in the conclusion section.
19.Does your paper have industrial applications? If yes, who are the likely users?
20.What is your main contribution to the field?
21.Finally, I would suggest the author to address the questions above in the revision. I am pleased to review the revised manuscript.
Reviewer 2 Report
The paper was revised according to the journal rules. The topic treated was focused on the removal of organic pollutants using TNA under solar illumination. Few revisions are required and they are reported below:
- please add the nomenclature section for all acronyms, parameters and abbreviation used
- I suggest to reduce acronyms from the abstract
- "but using ozone makes the process less environmentally benign" please add references to this aspect. I suggest also to clarify more this aspect of "environmental impact"
- the environmental impact of the such process should be considered
- check that all details for the instruments used in section 2 are clearly added
- being an experimental study I suggest to add an uncertainty analysis section for the measurements accomplished
- figure 1as is difficult to read
- the zoom in figure 1b should be enlarged
- the style of section 3.2 should be the same for all the manuscript
- ppm should per volume or per weight (ppm(v))
- add details for the decoloration efficiency
- in section 3.3 a tentative mechanism of this H2O2 effect could be added
- figures are quite blurry even if they are explicative
Reviewer 3 Report
In this paper, the authors introduced the study of a film-based ann TNA and H2O2 coupled system to degrade methyl orange, an azobenzene-type commercial textile dye, via a non-ferrous Fenton-type advanced oxidation process under solar illumination. The idea behind this is interesting. However, I still have quite a number of concerns in this manuscript. There are times where there are not enough data to support the conclusions of the author. Please see some of the major concerns below.
1.The information for the top surface of ann TNA is not clear. The authors should give much more information about this. So the readers can get its reproducibility. For, example the inside SEM image can’t be noticed in figure 1(a)
2. The authors should give much more information about the novelty of this paper, especially the effect of using this new study, which applications can be used this device?
3. It would be very useful to compare the performance of the present design with the state-of the-art of in the Introduction Section. A table comparing the performance of similar structures reported in literature needs to be included to enhance the quality of manuscript.
4. More references need to be included in the introduction part to understand the applications of using for device applications.
a. Nanostructures with periodic heating–cooling cycles for photoacoustic imaging using continuous-wave illumination- Journal of Nanophotonics, 2017
b. A Two-Channel Silicon Nitride Multimode Interference Coupler with Low Back Reflection- Applied Sciences, 2022
5. Much more discussion about the results should be given in this paper, especially the author needs to provide enough physicals mechanism analysis about the results.
6. The equations in the paper are not clear, the authors need to modified them, so readers can get its reproducibility
Reviewer 4 Report
Nanomaterials (ISSN 2079-4991)
Review of an article:
«Degradation of Organic Methyl Orange (MO) Dye Using Photocatalyzed Non-Ferrous Fenton Reaction»
by Sifani Zavahir, Tasneem Elmakki, Norhan Ismail, Mona Gulied, Hyunwoong Park, Dong Suk Han
In Nanomaterials (ISSN 2079-4991).
Round 1
The article is concerned with an experimental investigation of finding that a titanium nanotube array (TNA) based non-ferrous Fenton system could be used for the successful degradation of a model contaminant azo dye, methyl orange (MO), under simulated solar illumination. In this work, the non-ferrous Fenton performance of TNA in a minimal concentration of hydrogen peroxide showed remarkable degradation kinetics with a 99.7 % reduction in MO dye concentration after a 60 min reaction time when illuminated with simulated solar light (100 mW/cm2, AM 1.5G).
The article was presented in a well-structured manner, with a good level of organization. Unfortunately, several statements within have weak evidence. Therefore, the referee suggested that the manuscript be improved with a major revision. The following is a list of specific concerns.
1. Line 143: Please, add information concerning the time of experimental exposure (up to 60 min).
2. Figure 2b: There is no Y-axes scale. It is well-known, the visible light absorption spectrum for MO indicator has two maxima, and provides all range spectra from 190 to 650 nm. Also, specify the absorbance at 254 nm maxima to understand the possible formation of byproducts to prove TOC data.
- If this degradation is photochemically dependent and pH at the same range, the isosbestic point appears.
- Please, provide any data for clarification.
3. Reference list: There are only two works since 2022 in the reference list. Please, provide any valuable data to improve the quality of information from the literature.
Despite another type of composite, it is highly recommended to do the comparison with
- https://doi.org/10.1038/s41598-018-20569-w;
- https://www.mdpi.com/2073-4441/13/21/2948 etc.
4. The novelty of the work is unclear.
5. Please, enlarge the conclusion section and provide any information concerning future perspectives and outlooks.
6. Please, provide any LC/MS/MS chromatograms in the Supporting Information section.
Style guide:
· Line 1. Please, select the article type. It’s specified only in Susy.
· Line 144: It should be minus (–), not dash (-) etc.
English spelling should be double-checked.
Round 2
Reviewer 1 Report
The revised manuscript now can be accepted in the journal for publication as the authors have incorporated all the suggestions.
Author Response
Dear Reviewer, We greatly appreciate the constructive suggestions for improving our manuscript. Thank you.
Reviewer 3 Report
In this paper, the authors show the study of Removal of recalcitrant organic pollutants by degradation or mineralization from indus-trial waste streams is continuously being explored to find viable options on the commercial scale. Herein, we propose a titanium nanotube array (based non-ferrous Fenton system for the successful degradation of a model contaminant azo dye, methyl orange, under simulated solar illumination. titanium nanotube arrays were synthesized by anodizing titanium film in an electrolyte medium 15 containing water and ethylene glycol. Characterization by SEM, XRD, and profilometry confirmed 16 uniformly distributed tubular arrays with 100 nm width and 400 nm length.The idea behind this is interesting. However, I still have quite a number of concerns in this manuscript. There are times where there are not enough data to support the conclusions of the author. Please see some of the major concerns below.
1.The information for the SEM results is not enough. The authors should give much more information about this. So the readers can get its reproducibility.
2. The authors should give much more information about the novelty of this paper, especially the effect of using Removal of recalcitrant organic pollutants by degradation or mineralization from indus-trial waste streams, which applications can be used this device?
3. The fabrication tolerance analysis, which can offer a good guide for the fabrication requirement, and the key parameters, need to be added in the results section ?
4. More reference need to be included in the introduction part to understand the fabrication of nano-devices such as:
A photonic 1× 4 power splitter based on multimode interference in silicon–gallium-nitride slot waveguide structures
- Materials, 2016
Reviewer 4 Report
nanomaterials-2153675
Review of an article:
«Degradation of Organic Methyl Orange (MO) Dye Using Photocatalyzed Non-Ferrous Fenton Reaction»
by Sifani Zavahir, Tasneem Elmakki, Norhan Ismail, Mona Gulied, Hyunwoong Park, Dong Suk Han
In Nanomaterials (ISSN 2079-4991).
Round 2
The authors addressed well my comments. I would recommend its acceptance.
Author Response
Dear Reviewer, We thank you for your positive response to our paper.
Round 3
Reviewer 3 Report
The new version can be published.